# Atypical Centriolar Composition Correlates with Internal Fertilization in Fish

**DOI:** 10.3390/cells11050758

**Published:** 2022-02-22

**Authors:** Katerina Turner, Nisha Solanki, Hassan O. Salouha, Tomer Avidor-Reiss

**Affiliations:** 1Department of Biological Sciences, University of Toledo, Toledo, OH 43607, USA; katerina.turner@utoledo.edu (K.T.); nisha.solanki@rockets.utoledo.edu (N.S.); hassan.salouha@rockets.utoledo.edu (H.O.S.); 2Department of Urology, College of Medicine and Life Sciences, University of Toledo, Toledo, OH 43607, USA

**Keywords:** centriole, sperm, internal fertilization, external fertilization, sperm competition, evolution

## Abstract

The sperm competition theory, as proposed by Geoff Parker, predicts that sperm evolve through a cascade of changes. As an example, internal fertilization is followed by sperm morphology diversification. However, little is known about the evolution of internal sperm structures. The centriole has an ancient and evolutionarily conserved canonical structure with signature 9-fold, radially symmetric microtubules that form the cell’s centrosomes, cilia, and flagella. Most animal spermatozoa have two centrioles, one of which forms the spermatozoan flagellum. Both are delivered to the egg and constitute the embryo’s first two centrosomes. The spermatozoa of mammals and insects only have one recognizable centriole with a canonical structure. A second sperm centriole with an atypical structure was recently reported in both animal groups and which, prior to this, eluded discovery by standard techniques and criteria. Because the ancestors of both mammals and insects reproduced by internal fertilization, we hypothesized that the transition from two centrioles with canonical composition in ancestral sperm to an atypical centriolar composition characterized by only one canonical centriole evolved preferentially after internal fertilization. We examined fish because of the diversity of species available to test this hypothesis–as some species reproduce via internal and others via external fertilization–and because their spermatozoan ultrastructure has been extensively studied. Our literature search reports on 277 fish species. Species reported with atypical centriolar composition are specifically enriched among internal fertilizers compared to external fertilizers (7/34, 20.6% versus 2/243, 0.80%; *p* < 0.00001, odds ratio = 32.4) and represent phylogenetically unrelated fish. Atypical centrioles are present in the internal fertilizers of the subfamily Poeciliinae. Therefore, internally fertilizing fish preferentially and independently evolved spermatozoa with atypical centriolar composition multiple times, agreeing with Parker’s cascade theory.

## 1. Introduction

In his historical 1970 sperm competition paper, Dr. Geoff Parker discussed the sequence of possible events influencing sperm evolution [1]. On p.526, he says: “Thus it seems likely that the reproductive behavior of insects followed a path from external fertilization to internal fertilization with spermatophores, and later to copulation with the spermatophore deposited inside the reproductive tract of the female. A further stage has been reached in some groups with the introduction of free sperm transfer. Natural selection might have had a subordinate role to intrasexual selection during most of the earlier stages of this pathway.” This series of natural selection events is called the sexual cascade [2]. As part of this, internal fertilizers have evolved a large diversity of sperm morphologies [3,4] including longer sperm [5]. Additionally, significant attention has been given to evolutionary adaptions in sperm head and tail morphology [6,7,8,9,10,11]. However, little is known about the structural evolution of the sperm neck and the centrioles within it (Figure 1).

Centrioles have a distinct structure that has been maintained since the last eukaryotic common ancestor (LECA) about 1.5 billion years ago (Hodges et al., 2010). This structure is easily identifiable by the microtubule bundles organized into nine-fold symmetry, as observed by electron microscopy (Figure 1) [12,13,14,15]. While the most common and ancestral form consists of nine triplet microtubules, centrioles with nine doublet or nine singlet microtubules are also known to occur [16,17]. Based on these structural criteria, it was observed that many animal cell types including sperm cells have two centrioles, which is the ancestral centriolar composition of a sperm cell (Figure 1B). However, some exceptions were noted to have atypical centriolar composition. For example, both insect and mammalian sperm cells either have only one or no centrioles as defined by the above structural criteria [18,19,20,21,22,23,24]. Only recently, due to the discovery of the protein composition of the centriole, a second functional though structurally atypical centriole was observed first in insects [24,25,26,27,28] and later in mammals [22,29,30]. Suggesting that animal sperm cells indeed have two centrioles when they were thought to possess only one typical sperm centriole, even if they cannot be identified using classical structural criteria.

The two centrioles in animal sperm maintain distinct functions and locations relative to the nucleus. The centriole near the nucleus is called the proximal centriole, while the more distant centriole, which also nucleates the flagellum, is called the distal centriole [31]. Both centrioles form a centrosome post-fertilization in the zygote [21,32,33,34,35,36].

Centrioles are involved in several functions in somatic cells, particularly accurate cell division, ciliogenesis, and the synthesis of new centrioles [37,38,39,40]. However, in sperm, their main purpose is to form the tail (i.e., axoneme). Additionally, they have been proposed to regulate tail movement [29]. Consequently, it could be predicted that centrioles become more specialized (i.e., atypical) according to the environment that the sperm is moving through to maximize efficiency. For example, in sperm movement through water, which is the condition in external fertilization, centrioles may remain typical because water is a simple and homogenous environment, and sperm must swim for only a short time directly toward the eggs. However, when sperm travel through the female reproductive tract such as in internal fertilization, they may traverse more complex environments, triggering the next step in the sexual cascade [41,42]. Indeed, sperm swimming through water is a simple process relative to the process of internal fertilization [7].

The female reproductive tract is comprised of several environments that create distinct barriers for sperm travel or storage and may present more opportunities for sperm competition. In mammals, these barriers include the cervix, uterotubal junction (UTJ), and oviduct [43]. In insects, they include storage in specialized organs [44]. Additionally, the female reproductive tract has viscous mucus through which the sperm must move to reach the egg [45]. These properties of the female reproductive tract generate evolutionary pressure on sperm movement. We, therefore, hypothesized that internal fertilization in an elaborate female reproductive tract benefits from remodeled centrioles, whereas external fertilization in a simple water environment benefits from canonical centrioles.

During mammalian evolution, internal fertilization existed before the sperm centriole became atypical. Mammals were preceded by Synapsids, and most Synapsid groups (including birds [46], snakes [47], and turtles [48]) have ancestral centriolar composition with two canonical centrioles. Therefore, internal fertilization preceded the appearance of atypical centrioles in mammalian evolution.

Fish are among the most ancient vertebrate animals, are characterized by incredible diversity, and have a range of reproductive modes including internal and external fertilization. Fish sperm is commonly studied in decisions on the phylogenetic relationships of distinct species, and fish have a variety of sperm morphologies and structures [49,50]. It was noted that the sperm centrioles, like other sperm structures, could take different forms in fish. However, the relationship between centriolar structure and reproduction strategy is unclear. Here, we took advantage of this rich literature to determine the association of sperm centriolar composition with internal or external fertilization strategies. We found a strong and statistically significant correlation between internal fertilization and atypical centriolar composition, suggesting that atypical centrioles evolved as part of the evolutionary sexual cascade that occurred after the appearance of internal fertilization.

## 2. Materials and Methods

### 2.1. Fish Centriole Literature Search

We searched PubMed and Google Scholar for published electron microscopy studies of fish sperm and their centrioles using the search terms “fish, sperm, centrioles” and “fish, sperm, electron”. In total, we incorporated over 98 published documents into our study. We also searched two books by Jamieson that describe a systematic evaluation of sperm and included papers that may not otherwise be found by our search because they were not written in English or reported unpublished data [49,50].

### 2.2. Analysis of Sperm Centriole Number

Each paper was searched for claims on the presence and number of centrioles. We then confirmed these claims by examining the relevant paper’s figures using several known ultrastructural characteristics of the distal and proximal centrioles (Appendix A). Distal centriole (also referred to as basal body) presence was confirmed either when an electron-dense axoneme extension was present beyond the longitudinal canal in the neck in a longitudinal section; when nine triplet microtubules were seen in a cross-section at the axoneme base; or when we observed a ring of microtubules that lacked the central tubules characteristic of the axoneme. In some cases, the cross-section only showed an electron-dense ring or blurred triplets, presumably due to low image quality or an imperfect cross-section (oblique section). In one case, the cross-section showed nine doublets. Proximal centriole presence was confirmed when an electron-dense material resembling a longitudinal section of a barrel was present, when nine triplet microtubules were identified or when a ring was observed in the cross-section just below the nucleus. In some cases, we did not have access to the original image of the centriole and could not determine the strength of the evidence for the absence of a sperm centriole. In these cases, we marked the species as unconfirmed (Appendix A).

### 2.3. Statistics

We tested for a statistically significant difference using the Z-score calculator for two population proportions found at https://www.socscistatistics.com/tests/ztest/default2.aspx (accessed on 1 January 2022). The site used the equation:(1)$$\frac{\left({\bar{p}}_{1}-{\bar{p}}_{2}\right)-0}{\sqrt{\bar{p}\left(1-\bar{p}\right)\left(\frac{1}{n_{1}}+\frac{1}{n_{2}}\right)}}$$

### 2.4. Phylogeny

We used NCBI’s “Taxonomy” tools at https://www.ncbi.nlm.nih.gov/taxonomy (accessed on 1 January 2022) to determine the phylogeny of the fish species in this study.

### 2.5. Reproduction Mode

We determined the reproductive strategy of the various species by searching PubMed and Google Scholar using the species name. In cases where we did not find the specific reproduction strategy of a species, we looked for the reproduction strategy of related species in the same genus or family. The reproduction mode was also confirmed using Fishbase https://www.fishbase.se/search.php (accessed on 1 January 2022) [51].

## 3. Results

### 3.1. Most Fish Species Studied Ultrastructurally Are External Fertilizers

By searching the available literature that described electron microscopy studies of sperm, we surveyed reports on the sperm centrioles of 277 fish species (Appendix A, Table 1). This is a small fraction of the at least 34,000 extant species. Most species (or their close relatives) were external fertilizers (87.7%, 243/277), and few were internal fertilizers (12.3%, 34/277), a ratio of 7.1 to 1. This ratio is lower than estimated in nature, as it has been reported that more than 95% of fish species (ratio of more than 19:1) are external fertilizers [52], indicating a bias toward ultrastructural studies of internally fertilizing fish. 

Of the 277 species studied, 268 (96.8%, 268/277) were reported to have two canonical centrioles, while nine species (3.2%, 9/277) were reported to have only one canonical centriole. This is a ratio of 30 to 1, indicating that most fish studied were characterized by ancestral centriolar composition. 

### 3.2. Two Studied Externally Fertilizing Fish Species have Atypical Centriolar Composition

Of the 243 studied species that reproduce by external fertilization, two were reported to have one canonical centriole, representing 0.8% of the external fertilizers (Appendix A). These species reports are as follows: 

Gwo et al. reported on p.286 that in *Spratelloides gracilis,* “No proximal centriole was identified” [53], while it was reported that this species reproduces by external fertilization [54,55].

Fu et al. reported on p.60 that “the proximal centriole of *Engraulis japonicus* (Engraulididae) is indistinct” [56], and it was also reported to be an external fertilizer [51,57].

Many other externally fertilizing fish have two canonical centrioles. This includes fish from the families Scorpaenidae and Danioninae (which includes the important model species Zebrafish) [58,59]. For example, we studied papers on five species of Salmonidae (Table 1), all of which have two canonical centrioles [60,61,62,63,64].

### 3.3. A High Rate of Internally Fertilizing Fish Species Studied Have Atypical Centriolar Composition

Of the 34 species that reproduce by internal fertilization, seven (20.6%) were reported to have a single sperm centriole (Appendix A, Table 1). We used the Z test for two population proportions and found that internally fertilizing fish had a statistically significant higher proportion of sperm with a single canonical centriole than externally fertilizing fish (*p <* 0.00001). The relative rate reduction was 19.8% for the reported single canonical centriole. The odds ratio of 32.4 for the group reported with one canonical centriole over the group reported with two canonical centrioles indicates a high difference. A similar conclusion can be drawn if the data are calculated based on the number of genera studied instead of species name (Table 1). This finding suggests that the evolutionary sexual cascade of events resulted first in internal fertilization and later, in some species, in atypical centriolar composition.

### 3.4. Species of the Internally Fertilizing Fish Subfamily Poeciliinae Have Atypical Centrioles

Of all the fish analyzed, one group appears to have atypical centrioles: the subfamily Poeciliinae (belonging to Teleostei, or bony fishes with protrusible jaws) (Figure 2). Species of the subfamily Poeciliinae and the family Poeciliidae are internal fertilizers [65].

Jamieson reported on p.470 for *Poecilia reticulata* that “proximal centriole triplets become occluded and is reduced to a remnant by maturity” [49,50,66,67]. This species is an internal fertilizer [68].

Grier et al. reported on p.86 for *Poecilia latipinna* that “a rather indistinct electron dense structure (Figure 15) was present in the sperm neck” [69]. This species is an internal fertilizer [70].

Emel’yanova and Pavlov reported on p.97 that for *Gambusia affinis*, the “proximal centriole (pc) is electron dense and no longer contains distinct tubules” [71]. This species is an internal fertilizer [72].

The subfamily Poeciliinae includes many genera in addition to *Poecilia* and *Gambusia* that are thought to belong to distinct tribes [73]. Therefore, it is possible that the subfamily Poeciliinae is a monophyletic group that ancestrally had an atypical centriole.

### 3.5. Species with a Single Canonical Centriole Evolved Independently Multiple Times

In addition to the three studied species of the subfamily Poeciliinae, we identified four additional species with a single canonical spermatozoan centriole that are also internal fertilizers; each of them occupies a distinct phylogenetic clade (Figure 1). The reported internal fertilizers included one Chondrichthyes species (cartilaginous fishes) and three species from distantly related Teleostei (bony fishes with protrusible jaws) groups.

*Hydrolagus colliei* (Chondrichthyes): It was reported on p.193 that “the proximal centriole has not been seen in mature Hydrolagus sperm” [74], and this species was reported to be an internal fertilizer [75].

*Lepidogalaxias salamandroides:* It was reported on p.42 by Leung that “No typical 9 × 3 microtubule centrioles have been found in the spermatozoon, but a basal body was located at the anterior of the axoneme” [76], and this species was reported to be an internal fertilizer [77]. Jamieson reported on p.156 for *Lepidogalaxias salamandroides* that “no typical triplet centrioles have been observed. A basal body…located at the anterior end of the flagellum consists of 9 doublets” [49].

*Hemirhamphodon pogonognathus*: It was reported on p.251 that “A proximal centriole has not been identified” [78] and this species was reported to be an internal fertilizer [79].

*Pantodon buchholzi*: It was reported on p.118 that “the centriolar complex consists of a single centriole (basal body), although an additional centriole, presumably the proximal centriole, is sometimes observed parallel to it” [49]. This species was reported to be an internal fertilizer [49,75].

Additionally, the two reported external fertilizers were from two unrelated Teleostei groups. *Spratelloides gracilis* (silver-stripe round herring) and *Sardinops melanostictus* both belong to the Clupeoidei suborder, though they are from distinct families (Engraulidae or Anchovies and Clupeidae). The Clupeidae family includes the species *Sardinops melanostictus*, whose spermatozoa have two canonical centrioles [49]. Therefore, the appearance of atypical centriolar composition in the Clupeoidei suborder occurred twice independently.

The nine species with a single canonical centriole belonged to clades that include many more species with two canonical centrioles, suggesting that ancestrally, they had spermatozoa with two canonical centrioles (Figure 2). Therefore, it appears that a single canonical centriole is a recent innovation in the evolution of these fish species, and that a change from ancestral centriolar composition to atypical centriolar composition took place independently multiple times, indicating convergent evolution due to positive selection.

### 3.6. The Atypical Centriole Forms during Spermiogenesis

In many animal species, the canonical structures of the proximal and distal centrioles are maintained during spermiogenesis (the differentiation of a round spermatid to a spermatozoon in the testes) [12,80]. Similarly, the canonical structures of the proximal and distal centrioles are maintained during spermatogenesis in fish with ancestral centriolar composition [81]. In mammals and insects, the atypical centriole forms during spermiogenesis via distinct mechanisms [21,23,24,82]. In insects, an atypical proximal centriole forms during spermiogenesis as a new centriole. In mammals, the canonical distal centriole found in the early spermatid is remodeled during spermiogenesis to become atypical. This indicates that insects and mammals use very different mechanisms to create the atypical centrioles during spermiogenesis.

Spermiogenesis in the fish *Gambusia affinis,* which has atypical centrioles, was studied [69]. In this species, the canonical proximal and distal centrioles are present in the early spermatid differentiation stage, but the proximal centriole becomes occluded in the later spermatid stage, and its microtubules disappear by the end of spermiogenesis. This situation is like that of insects, in that their atypical sperm centriole is the PC, not the DC, as is the case in mammals. However, *Gambusia affinis* is like mammals in that its atypical sperm centriole is remodeled from a spermatid centriole and not formed initially as an atypical centriole in the spermatid, as it is in insects. It therefore appears that atypical centriolar composition is achieved by distinct subcellular mechanisms in mammals, insects, and fish, which is consistent with the idea that they all evolved independently.

## 4. Discussion

By reviewing the literature on the sperm centriolar structure of 277 fish species, we made two observations:the presence of only one recognizable centriole is continually (20.6% of internal fertilizers studied) and specifically (*p* < 0.00001 when comparing internal to external fertilizers) associated with internal fertilization; andthe presence of only one recognizable centriole evolved independently at least six times during fish evolution. This is in addition to the two independent incidents of only one canonical centriole evolving in insects and mammals. These data are consistent with Parker’s evolutionary sexual cascade theory.

In insects and mammals that were extensively investigated, it was found that the sperm have, in addition to the one recognizable canonical centriole, a second, atypical centriole, which was recognized only recently due to the use of centriole-specific protein markers [22,24,27,30,83]. Therefore, we speculate that some fish with one recognizable centriole will also have a second centriole that is not readily recognizable. This speculation will need to be investigated further.

We found that two species of external fertilizers only had one recognizable centriole. Possible explanations for this observation are: (i) they may have evolved from a species that was an internal fertilizer in the past and became external fertilizers as a secondary trait; and (ii) the sperm of these species or their ancestors may have been released to a complex external environment that has some similarity to the complex conditions of the female reproductive tract.

One possible explanation for the evolution of atypical sperm centriolar composition is the significantly higher rates of multiple paternity in internally fertilizing fish species [52]. Additionally, most fish that exhibit complex patterns of parental behavior are freshwater forms [84]. Interestingly, we found that six of the seven internal fertilizers with one canonical centriole (*Poecilia reticulata*, *Poecilia latipinna*, *Gambusia affinis*, *Lepidogalaxias Salamandroides*, *Hemirhamphodon pogonognathus*, and *Pantodon buchholzi*) live in freshwater. It is tempting to speculate that the atypical centriole trait is another strategy used, along with complex patterns of parental behavior, to gain an advantage during sperm competition.

Of the above species, *Poecilia* species are used as a common reproductive model. Guppies and mollies are used to study sexual conflict [85], sperm competition [86], the impact of predation on sperm [87], and many other topics [88]. It would be important to study the precise structure of the atypical centriole in these species using more advanced electron microscopy and to study how the structure evolved in this group. It would also be useful to compare the protein composition of canonical and atypical sperm centrioles in fish and to extend studies of sperm centrioles to other fish groups. For example, the spermatozoa of triploid gynogenetic crucian carp (*Carassius auratus*) have abnormal expression of centriole-related genes [89], while zebrafish is a widely used fish model with two canonical centrioles that are essential for normal embryo development [90,91].

A key question is why does the sperm centriole become atypical in internal fertilizers? A clue may come from the centriole’s strategic location in the sperm neck, which connects the sperm head and tail. Reproductive success depends on efficient sperm movement driven by axonemal, dynein-mediated microtubule sliding. Some models predict sliding to occur at the base of the tail (i.e., the atypical centriole) [92]. Canonical centrioles are rigid structures that restrict microtubule sliding. We found that, in bovine sperm, the atypical distal centriole, the canonical proximal centriole, and their surrounding atypical pericentriolar matrix form a dynamic basal complex that facilitates a cascade of internal sliding deformations that couple tail beating with asymmetric head kinking [29]. During asymmetric tail beating, the atypical centriole’s right side and the area surrounding it slide ~300 nm rostrally relative to the left side. The resulting deformation throughout the dynamic basal complex is transmitted to the head–tail junction. Thus, when the tail beats to the left, the head tilts to the left, generating a head kinking motion. These findings suggest that the dynamic basal complex evolved as a dynamic linker that couples the sperm head and tail into a single, coordinated system.

In fish and insects, whose atypical centriole is the PC, it is less clear what can be gained from it becoming atypical. One possibility is that the atypical PC occupies less space in a very crowded, narrow location such as the cytoplasmic invagination just below the nucleus of *Tribolium castaneum* [24]. Another possibility is that the atypical PC provides a different way to couple sperm tail movement to the head, potentially by working with other neck structures that extend to the tail. For example, the *Drosophilla melanogaster* PCL, which is found above the mitochondria derivative, may interact with it to transmit movement in the sperm tail to the head [23].

## 5. Conclusions

Atypical centriolar composition has evolved independently at least eight times during animal evolution, suggesting a strong positive selection. In six of the eight instances, centriolar composition changed with internal fertilization clades, suggesting that atypical centrioles tend to evolve in association with internal fertilization as part of the evolutionary sexual cascade. Further research is needed to characterize the fish sperm atypical centriolar structure, composition, and fate post-fertilization in the zygote.

## Figures and Tables

**Figure 1 cells-11-00758-f001:**
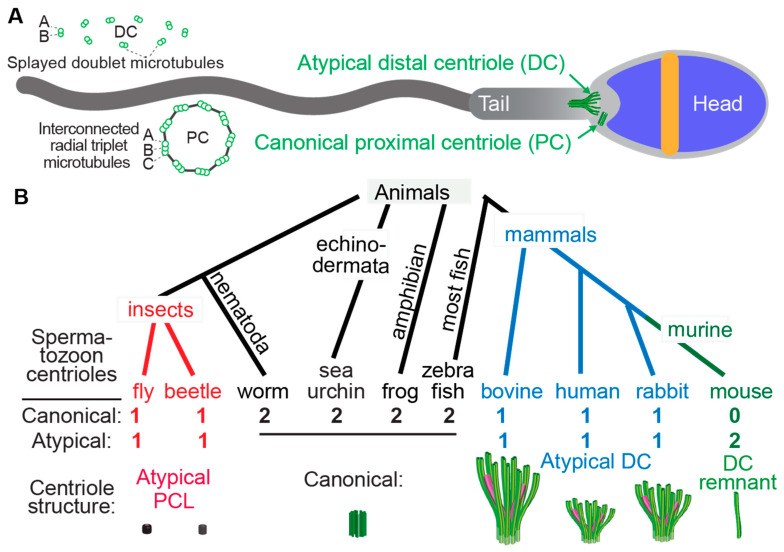
Atypical centrioles have distinct structures. (**A**) A mammalian sperm showing the locations of the atypical and canonical centrioles in the sperm neck. To the left, a cross section of the PC and DC depicting microtubule organization. (**B**) The number and size of spermatozoan centrioles vary throughout evolution. Shown are four animal groups (color-coded) organized based on the number and type of their spermatozoan centrioles. Animals with two centrioles (black). Animals with just one canonical centriole also have an atypical centriole: the proximal centriole-like (PCL, red) structure in insects and the atypical distal centriole (DC, blue) in mammals. Animals with DC remanent (green). Color code in the Atypical DC: green color marks the presence of microtubules, and magenta marks rod proteins.

**Figure 2 cells-11-00758-f002:**
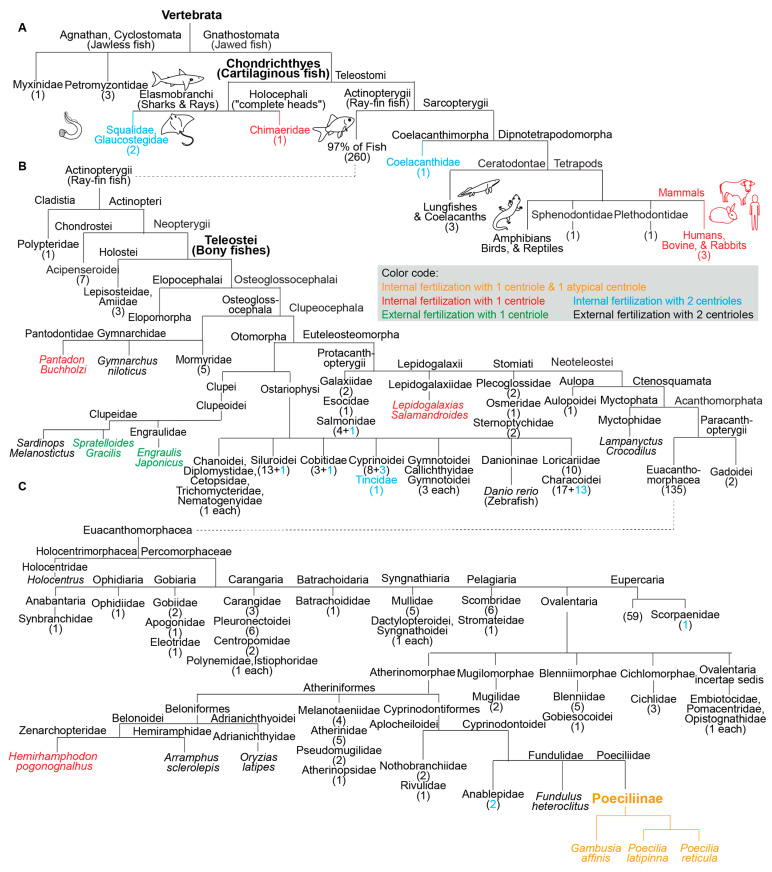
Fish species with a single canonical centriole evolved independently multiple times from an ancestral state with two canonical centrioles. (**A**–**C**) A phylogenetic tree depicting fertilization mode (internal or external) and canonical centriole number (1 or 2) in the main branches of vertebrates and fish. (**A**) Depicts the main fish groups. (**B**) Depicts ray-fin fish (Actinopterygii). (**C**) Depicts Euacanthomorphacea. In some cases, we provide the scientific name and common name in parentheses. Vertebrata, Chondrichthyes (Cartilaginous fish), Teleostei (Bony fishes), and Poeciliinae are bolded and enlarged because they are specifically referenced in the text. The number of fish species in a category appear as a numeral in parentheses. The figure contains the names of all fish families analyzed in this survey.

**Table 1 cells-11-00758-t001:** Number and percent of species and genera characterized by external or internal fertilization and reported to either have a single canonical centriole or two canonical centrioles.

Fertilization	One Centriole	Two Centrioles	Total
External	2 species (0.8%), 2 genera (1%) *Spratelloides gracilis* (Silver-stripe round herring) *Engraulis japonicus* (Japanese anchovy)	241 species (99.2%) 199 genera (99%)	243 species (87.7%) 201 genera (86.6%)
Internal	7 species (20.6%), 6 genera (19.4%) *Hydrolagus colliei* (Spotted ratfish) *Lepidogalaxias Salamandroides* (Salamanderfish) *Hemirhamphodon pogonognathus* *Pantodon buchholzi* (Freshwater butterflyfish) 3 Poeciliinae Species: *Poecilia reticulata, Poecilia* *latipinna* Guppy, and *Gambusia affinis* (Western mosquitofish)	27 species (79.4%) 25 genera (80.6%)	34 species (12.3%) 31 genera (13.4)
Total	9 species (3.2%) 8 genera (3.4%)	268 species (96.8%) 224 genera (96.6%)	277 species 232 genera
*p*-value	Species: <0.00001 Genera: <0.00001		
Ratio	Species: 3.5 (7/2) Genera: 3 (6/2)	Species: 11.2 (27/241) Genera: 12.6 (25/199)	
RRR	Species: 20.6%−0.8% = 19.8% Genera: 19.4%−1% = 18.4%		
Odds ratio	Species: (7/27)/(2/241) = 0.259/0.008 = 32.4 Genera: (6/25)/(2/1.99) = 0.24/0.01 = 24		

A Z-score calculator was used to determine the *p*-value for two population proportions. The species name is provided for those with a single canonical sperm centriole. RRR, Relative rate reduction.

## Data Availability

Not applicable.

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
