# Peer review of "Atypical Centriolar Composition Correlates with Internal Fertilization in Fish"

_cells, 2022, doi:10.3390/cells11050758_

Round 1

Reviewer 1 Report

            This report is a meta-analysis to study the appearance of centriole structure in sperm of fishes. It is purported to be a study of the evolution of single versus double centrioles in sperm and they conclude that internally fertilizing fish preferentially evolved sperm with only one centriole. This conclusion is based on a survey of reports of 277 fish out the at least 34,000 species of fish out there.  I am not convinced that this small sample size is adequate to back the very firm conclusion made in the manuscript. If anything, they have found a correlation that might be studied further and the title should reflect such. What is also lacking is a cogent discussion of why such a potential evolutionary change might be an advantage.

            In terms of specific points to raise. I only suggest that they cite the more original foundational research on sperm centrioles rather than the self citations. The work, analysis and presentation are all well done and adequately presented.

Author Response

We greatly appreciate the reviewers’ comments and that s/he recognizes that the work, analysis, and presentation are well done and adequately presented.

We have added to the paper that we surveyed reports on the sperm centrioles of 277 fish and that this is a small fraction of the at least 34,000 extant species.

We have changed the title to indicate that we report a correlation.

We have added a discussion of why an atypical centriole might be an advantage.

We have added citations of the original papers reported in our cited review papers.

Reviewer 2 Report

The authors compared 277 fish species spermatozoa centriolar structure,and suggested internally fertilizing fish preferentially and independently evolved spermatozoa with atypical centriolar composition multiple times. The results provided some useful information, but I they just compared the subfamily Poeciliinnae lacking other families’ results such as Scorpaenidae. In addition, the results did not provide new and innovative findings.

Author Response

We greatly appreciate the reviewer's suggestions and recognition that the results provided useful information.

In the main text and Fig 2, we added information on externally fertilizing families such as Scorpaenidae.

We added Scorpaenidae, and the family names of all other fish studied in this work, to Fig 1.

Reviewer 3 Report

In general terms, it is an interesting work that performs an adequate search for the information associated with centrioles in the sperm of animals that maintain different functions and locations in relation to the nucleus. But it is necessary to incorporate some minor details for possible publication.

Introduction.

Line 53-69: In this paragraph it is important to include a figure that considers differences and similarities between centrioles in different animals (for example: terrestrial animals, aquatic animals, amphibians and insects).

Results:
line 230: Are there studies on salmonidae? If so, can you provide some background information?

Discussion:
Line 304. Incorporate the contributions of this review to the scientific development associated with the biology of animal reproduction in different study models in fish.

Author Response

In general terms, it is an interesting work that performs an adequate search for the information associated with centrioles in the sperm of animals that maintain different functions and locations in relation to the nucleus. But it is necessary to incorporate some minor details for possible publication.

  • We greatly appreciate the reviewer’s suggestions and recognition that the manuscript is an interesting work with an adequate search.

Introduction.

Line 53-69: In this paragraph it is important to include a figure that considers differences and similarities between centrioles in different animals (for example: terrestrial animals, aquatic animals, amphibians and insects).

  • As suggested, we add a new figure (Fig 1) that depicts differences and similarities between centrioles of different animals

Results:

line 230: Are there studies on salmonidae? If so, can you provide some background information?

  • We added Salmonidae, the other standard model fish (Zebrafish), and the family names of all other fish studied in this work.

Discussion:

Line 304. Incorporate the contributions of this review to the scientific development associated with the biology of animal reproduction in different study models in fish.

  • We incorporated the contributions of this review to the more general scientific development associated with fish biology in the discussion.

Round 2

Reviewer 2 Report

The revised version is much better.